# Fifteenth century CE Bolivian maize reveals genetic affinities with ancient Peruvian maize

Huan Chen[1,2,3], Amy Baetsen-Young[4], Addie Thompson[1,3,4,5,6], Brad Day[1,3,4], Thelma Madzima[1,2,3,5], Sally Wasef[7], Claudia Rivera Casanovas[8], William Lovis[9]*, Gabriel Wrobel[9]*

[1]Genetics and Genome Sciences Program, Michigan State University, East Lansing, United States; [2]Department of Plant Biology, Michigan State University, East Lansing, United States; [3]Molecular Plant Sciences Program, Michigan State University, East Lansing, United States; [4]Department of Plant, Soil and Microbial Sciences, Michigan State University, East Lansing, United States; [5]Plant Breeding, Genetics and Biotechnology Program, Michigan State University, East Lansing, United States; [6]Plant Resilience Institute, Michigan State University, East Lansing, United States; [7]Genomics Research Centre, Queensland University of Technology, Brisbane, Australia; [8]Universidad Mayor de San Andres, La Paz, Bolivia; [9]Department of Anthropology, Michigan State University, East Lansing, United States

*For correspondence:
lovis@msu.edu (WL);
wrobelg@msu.edu (GW)

Competing interest: The authors declare that no competing interests exist.

## eLife Assessment

This **useful** study attempts to place an ancient maize sample from Bolivia, dated to the end of the Incan empire, in genetic and geographical context. The analyses show that this sample is most closely related to ancient Peruvian maize, but the data remain **inadequate** to determine the direction of dispersal and the extent of Inca influence over the genetic make up of the analyzed sample. There are additional deficiencies in the statistical analyses and selection inferences. The topic of the study would appeal to researchers studying maize dispersal and adaptation.

**Abstract** Previous archaeological and anthropological studies have demonstrated the myriad of ways that cultural and political systems shape access to food and food preferences. However, few studies have conducted a biocultural analysis linking specific genotypic/phenotypic traits as evidence of cultural selection in ancient contexts. Here, we provide insight into this topic through ancient genome data from Bolivian maize dating to ~500–600 BP, included as an offering with the mummified remains of a young girl. These data are compared to 16 previously published archaeological maize samples spanning at least 5000 years of evolution, and 226 modern maize samples. Our phylogenetic analysis showed that the archaeological Bolivian maize (aBM) has the closest genetic distance to the archaeological maize from ancient Peru, which in turn shared the most similarities with archaeological Peruvian maize. During the period of interaction between the Inca state and local polities in the central Andes and consequent interactions with local agricultural traditions, the genetic diversity of maize increased. Ovule development in modern maize was selected and compared to those in archaeological specimens, revealing evidence of targeted breeding strategies aimed at improving seed quality and yield. While the cultural origin of the maize – either Inca or local Aymara – is uncertain, we demonstrate that the samples are most similar to Peruvian maize and potential targeted selection strategies for enhanced growth were well established by the 15[th] century.

## Introduction

Modern maize (*Zea mays* ssp. *mays L.*) is one of the most important and productive global food crops (*Ranum et al., 2014*). Prior to European contact in the Americas, maize had been adapted for a wide range of environments. Its versatility is clear from its extensive geographic distribution today, ranging from 58° north latitude to 40° south latitude, and growing below sea level as well as 3600 meters above sea level in the Andes (*Bonavia, 2013*). In Bolivia, there is empirical evidence that small maize varieties can grow at altitudes of 3800–3900 meters above sea level (masl) in some areas of the Lake Titicaca Basin (*Chavez, 2006*), as well as in other sheltered areas of Oruro and Potosí, although with lower levels of production (*Hastorf et al., 2006*; *Suca Gómez, 2011*). Studies of archaeological samples and modern molecular data indicate that maize originally evolved in Mexico from the flowering species of grass, teosinte (*Zea mays* ssp. *parviglumus*) ~9000 years before the present (cal. BP) (*Doebley, 2004*; *Matsuoka et al., 2002*) and spread into South America by ~7000 cal BP *Grobman et al., 2012Lombardo et al., 2020* from the lowland and Eastern North America by ~2500 cal BP (*Hart and Lovis, 2013*). During this domestication process, there was hybridization between maize and mexicana (*Zea mays* ssp. *mexicana*) (*Hufford et al., 2013*). The dispersal of maize to the rest of the world followed European colonization of the Americas in the 15[th] and 16[th] centuries AD (*Bonavia, 2013*; *Rebourg et al., 2003*). The evolution and diversity of maize has been shaped by both cultural and natural selection in a wide range of environmental and social contexts. For example, previous research has shown that temperate adaption in modern maize is characterized by early flowering (*Swarts et al., 2017*), evidence has also shown that cultural preferences were a factor in maintaining otherwise undesirable traits (*Bellon, 1996*; *Knapp and Zimmerer, 1997*). Maize seeds were selected and retained by farmers for use in subsequent growing years to maintain or enhance specific morphological traits, such as color and texture, based on changing cultural preferences, sometimes at the cost of reduced cultivability and/or nutrition (*Louette and Smale, 2000*).

Politics and culture have had a strong influence on crop planting and food preferences in ancient (and modern) civilizations. Indeed, in the Andes, previous research showed that under the Inca empire, maize was fulfilled multiple contextual roles. In some cases, it operated as a sacred crop (*Mintz, 1985*), symbolic of political power and tightly associated with high-status or elite social groups (*Doebley, 2004*; *Bray, 2003*). It was also an important element in numerous cultural events and rituals (*Ceruti, 2015*). From an agriculture-economic point of view, planting a crop on the most suitable land takes priority if a farmer wishes to maximize the output-to-input ratio. However, in certain civilizations, including Andean groups, maize may not necessarily have been a crop for which yield was considered a deciding factor. For example, in Andean highland communities, maize was not the primary food crop during pre-Inca times (*Burger and Van Der Merwe, 1990*), not only because maize was ill-adapted to arid highland conditions and its cultivation was risky during the short growing season, but also because planting maize did not yield a return on investment. In the Andes during the Spanish conquest, the chronicler Cobo observed that, 'Indians dropped all that they were doing in order to produce a little maize, even though it cost them more than it was worth.' It seemed to him that work in the fields was '…one of the major forms of recreation and festivals…,' and he was impressed by the 'extraordinary fondness for farming' (*Doutriaux, 2001*; *Cobo, 1990*). While Indigenous agriculture and foodways would have been far more nuanced than Spanish chroniclers would have appreciated, this supports the fact that Andean farmers were not deterred from planting an abundance of maize in sub-optimal environments. In fact, while it was a desirable food, it could also have acted as a symbol of empire (*Mintz, 1985*) and a ritual-feast crop, since its first introduction to the region around 2500 BCE (*Pearsall, 1978*).

The current study contributes to broader investigations documenting maize biogeographic patterns in South America by focusing on maize samples from a 15[th] century elite stone tomb burial context from south of La Paz, Plurinational State of Bolivia. We sequenced two aBM kernels with six libraries, using short read sequencing. In parallel, we used [14]C Accelerator Mass Spectrometer (AMS) radiocarbon dating to determine the approximate age of the samples. Phylogenetic analysis of aBM, 16 archaeological and 226 modern maize samples (*Zea mays* ssp. *mays L.*), 88 modern samples of Teosinte (*Zea mays* ssp. *parviglumis*), 79 modern samples of Mexicana teosinte (*Zea mays* ssp. *mexicana*) and 1 *Tripsacum* genotype revealed genetic relationships among themselves (*Supplementary file 1, table S1*; *Figure 6—source data 1*). Finally, the unique single-nucleotide polymorphisms (SNPs) in aBM infer that the selection imposed to enhance reproductive success in marginal highland

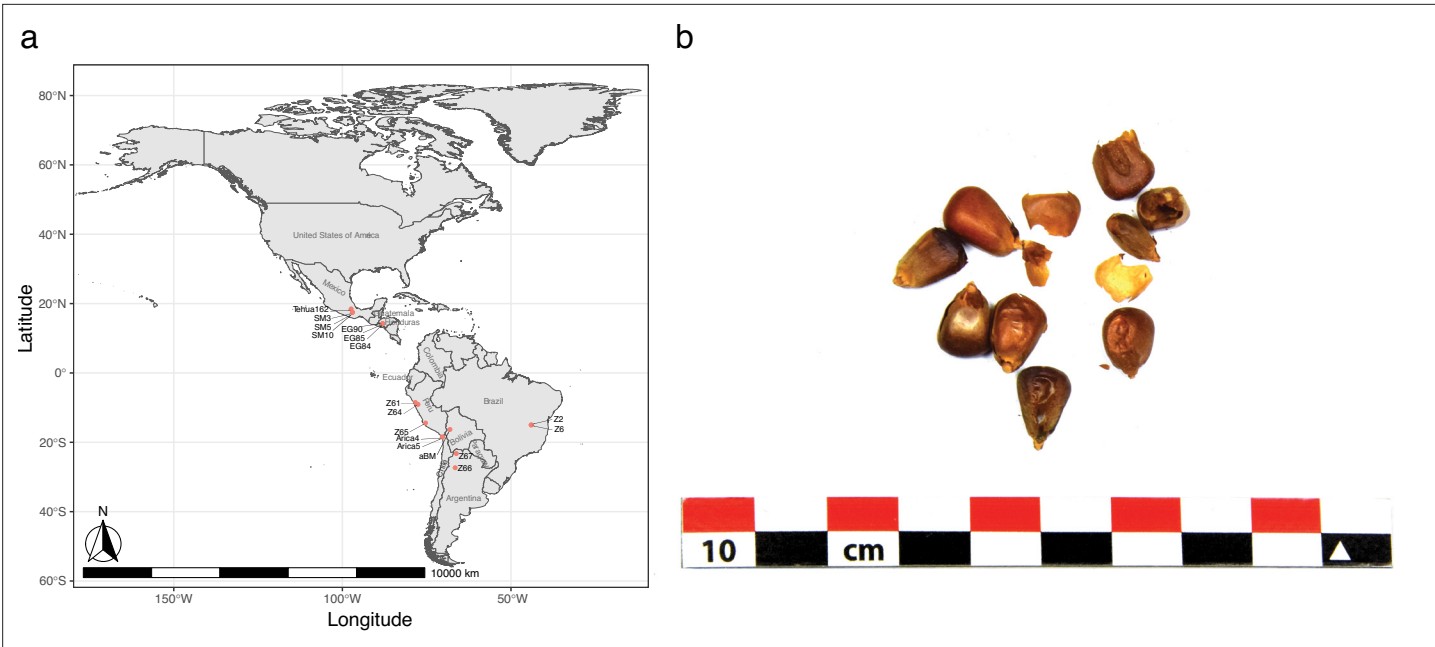

**Figure 1.** Location and appearance of archaeological Bolivian maize (aBM). (**a**) Map of maize sample collection and location information. (**b**) Photographs of kernels showing morphological characteristics.

growing environments resulted in increased maize diversity. Based on the data presented herein, we conclude that a combination the social and political needs of the Inca and local ethnic groups, and mitigation of certain natural environmental factors may have contributed to the specific preference for, and diversification of maize in the Central Andes. As such, our conclusions offer insight into the possible pathways that influenced maize genetic diversity, thus providing a vital resource to further investigate the factors that drove selection for maize domestication and diversity at the cusp of the cultural influence of the Inca empire.

## Results

### aBM sample

La Paz is a major city located in the highlands of west-central modern Bolivia, which is 16°30′S, 68°09′W, 3,650 masl. Dry conditions in this region and the adjacent Andes Mountain range have aided in the preservation of desiccated organic material. Our maize samples were found within a woven camelid fiber pouch that was one of many offerings associated with the mummified remains of a child reported as being found in a *chullpa* (stone tower tomb) south of La Paz, Plurinational State of Bolivia around 1890 (*Figure 1*). Although it is impossible to know the exact location of the *chullpa* or the origin of the individual, the assemblage was acquired by the U.S. Consul to Chile and was sent as a gift to the Michigan State University Museum, where she was curated until 2019. Following a series of recent analyses, she has since been voluntarily repatriated to Bolivia (*Lovis, 2023*). Given the reported archaeological context of the mummy and her abundant offerings recorded in the MSU Museum intake records, it has been proposed that she might represent an elite member of society. However, the nature of the coarse fabrics, simple design motifs, and restricted color palette of her textiles would suggest a more common individual, perhaps with an affiliation with a local high-status group. The vast majority of the funerary towers that exist in a radius of 60 km south of La Paz city are made of mudbricks. To our knowledge, the only exception is an Inca 'cushioned' (i.e. rounded edges) stone and mudbrick tower located in Uypaca, in the nearby Achocalla Valley, 8 km southwest of the city center (*Lovis, 2023*). Because of its characteristics, this tomb most likely belonged to a local (possibly non-Inca) high-status group and was part of a more extensive funerary site. Although it is impossible to know the exact provenience of the mummy, it would not be unreasonable to suppose that she originated from this or another similar tomb in the Achocalla Valley. Regardless, in the decades following her death,

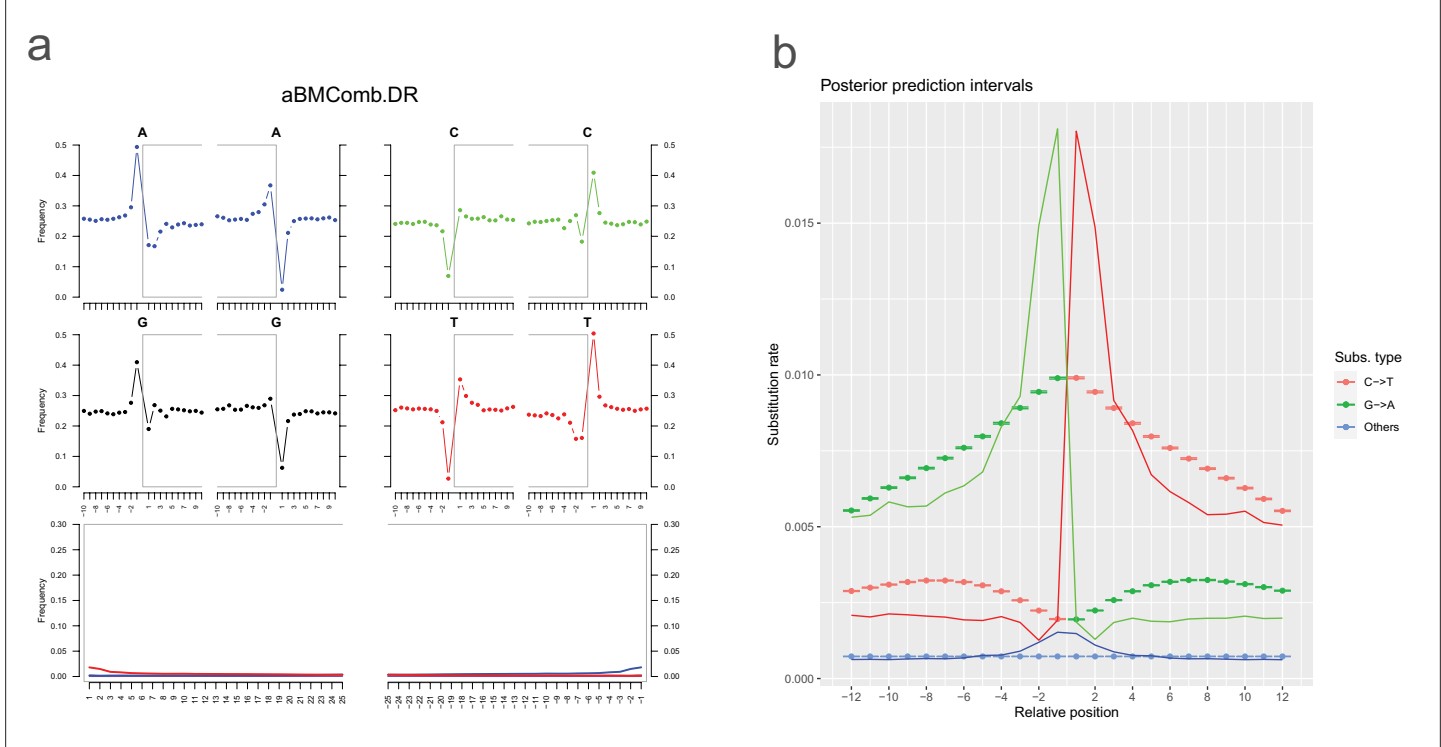

**Figure 2.** Ancient DNA (aDNA) damage pattern of archaeological Bolivian maize (aBM). (**a**) Cytosine deamination damage patterns for the combined BAM file of six aBM sequence samples and individual six aBM sequence samples. The position-specific substitutions from the 5' end (red) and the 3' end (green) of a read. The red line corresponds to C to T substitutions, the green line corresponds to G to A substitutions, and the blue line represents other types of substitutions. (**b**) Ancient DNA damage profile. The four upper plots show the base frequencies inside and outside of a read, where the open gray box corresponds to a read. The two lower plots show the position-specific substitutions from the 5' end and the 3' end of a read. The red line corresponds to C to T substitutions, the blue line corresponds to G to A substitutions, and the fade line represents other types of substitutions.

community members periodically visited the *chullpa* and left offerings. These included the pouch full of well-preserved maize from which we collected and analyzed samples from two kernels.

A single Accelerator Mass Spectrometer (AMS) date was obtained from one kernel, yielding a radiocarbon age of 400±26 BP (D-AMS-027148; Zea mays kernel; corrected for fractionation, no $\delta^{13}C$ reported). For the date 400±26 BP, the two possible calibrated age ranges are 1455–1516 cal AD ($p=0.489$) and 1538–1627 cal AD ($p=0.511$), with a median age of cal AD 1534 (Calibrated at 2σ with Calib 8.2 [SHCal 20.14 c]). Choice of the earlier age range is supported by other absolute dates on the assemblage (*Lovis, 2019*; *Hogg et al., 2020*): 447±45 BP (D-AMS-32518; Cucurbitaceae, $\delta^{13}C=-27.6$) with two possible age ranges of 1443–1504 cal AD ($p=0.848$) and 1594–1616 cal AD (0.152) with a median age of 1474 cal AD; and 510±20 (PSUAMS-5947; Camelidae fiber, $\delta^{13}C=-18.9$) calibrated to 1420–1454 cal AD ($p=1.00$) with a median age of 1438 (Both samples calibrated for southern hemisphere at 2σ with Calib 8.2 [SHCal 20.14 c]). Ages on the squash and maize are statistically identical at $p=0.95$ (t=1.70, $x^2.05=3.84$, df=1) with a pooled mean of the two ages of 424±18 BP calibrated to 1453–1505 cal AD ($p=0.769$) and 1593–1617 ($p=0.231$). Results on the squash, with the highest probability between 1443–1504 cal AD, and camelid fibers with an age range of 1420–1454, as well as the pooled mean, reveal a greatest likelihood for a mid- to late 15th c age for the assemblage, consistent with the earlier of the two calibrated age ranges on the maize kernel.

Two associated maize kernels from the same pouch were subjected to molecular analysis. Genomic base modification patterns associated with ancient DNA (aDNA), including cytosine deamination damage patterns, have been identified. Observed position-specific substitutions at the ends of the sequence reads (e.g. C→T/G→A *Handt et al., 1996*; *Krings et al., 1997*; *Hansen et al., 2001*; *Gilbert et al., 2003*; *Dabney et al., 2013*; *Figure 2a*), as well as an increase in adenine (A) and guanine (G) residues at the 5' end, concomitant with a reduction of C and T residues (*Dabney et al., 2013*; *Briggs et al., 2007*; *Stiller et al., 2006*; *Figure 2b*). From this, we evaluated all substitution types and read

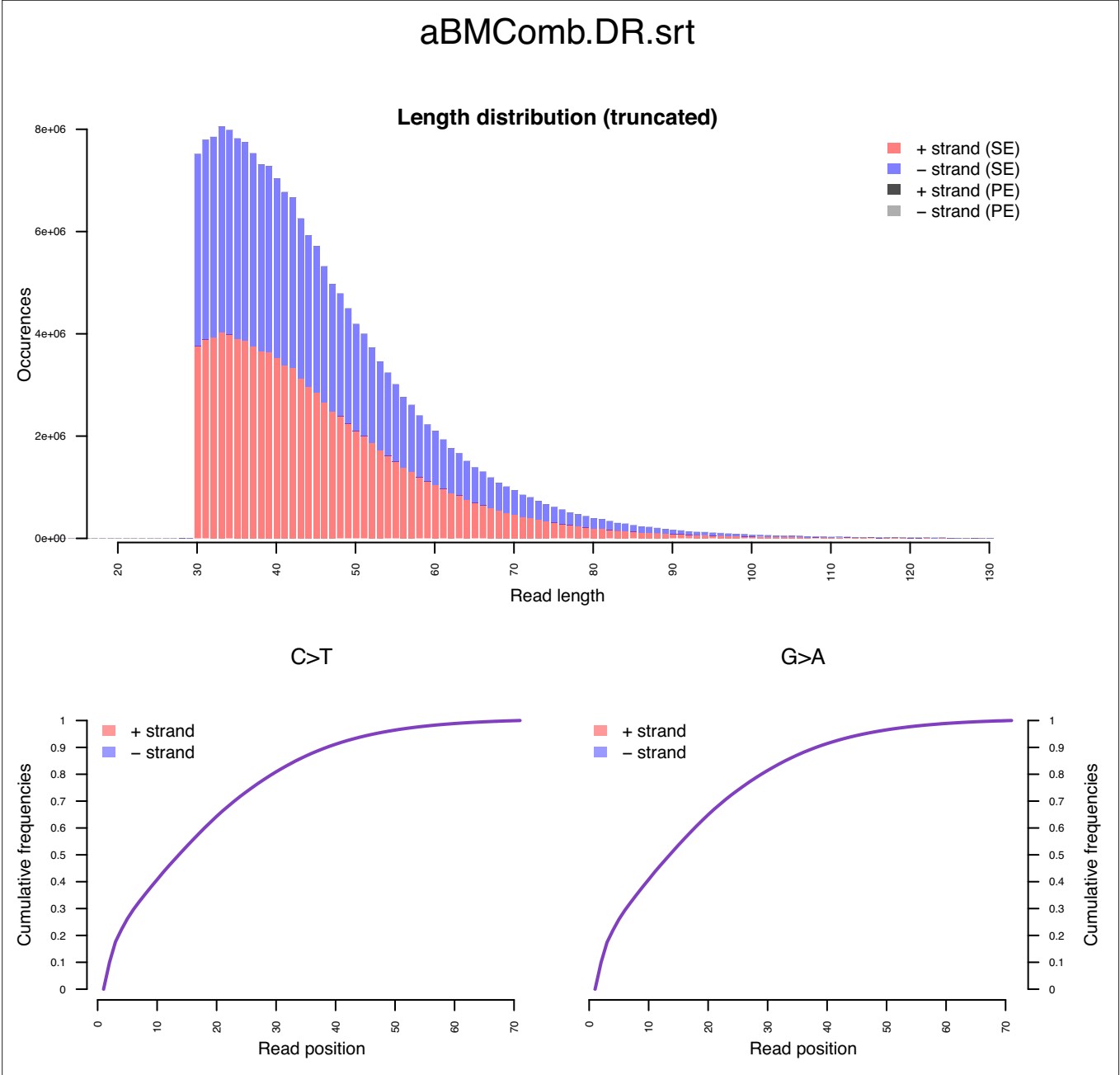

**Figure 3.** Archaeological Bolivian maize (aBM) DNA read length. (**a**) Read length distribution of combined aBM bam files prior to hard-masked 5′ thymine and 3′ adenine residues within 5 nt of both ends.

length (*Figure 3*), including read length range (*Pääbo, 1989*) and substitution rate types, which are associated with aDNA genome data (*Prüfer et al., 2010*). In total, these data are consistent with the radiocarbon results, supporting that (1) with the maize is in fact consistent with an archaeological biological sample, and (2) with based on the geographic origin of the sample, the kernels potentially descended from the period of the last Inca empire before the Spaniard Francisco Pizarro's conquest of the Inca capital, Cuzco, in A.D. 1533 (*Wilson et al., 2007*).

Our study focused on identifying the location from which aBM originally came, establishing and explaining patterns of genetic variability of maize, with a specific focus on maize strains that are related to our current aBM samples. We were especially interested in exploring how these patterns of variability found in both archaeological and modern samples reflect the influence of cultural systems

of behavior on both the spread of variants across the ancient landscape and selection for phenotypic traits by local populations.

## Genetic identity and genetic relatedness

After Tupac Inca Yupanqui became the emperor of the Inca Empire in the 15th century, the territory of the Inca Empire was greatly expanded across much of western South America, including the Qolla-suyu, a broad territory located in what is now western Bolivia, southern Perú, northern Chile, and non-highland Argentina (*D'Altroy, 2003*). Different ethnic groups populated this territory forming polities that were confederated at the time of the Inca arrival. Several Aymara Pakaje groups inhabited the highlands and the inter-Andean valleys south of La Paz, from where the maize samples derive (*Wilson, 1999*). Because of maize's central role in Inca society—as both a staple crop and a symbolically charged resource among diverse ethnic groups under Inca influence—its widespread distribution may reflect the expansion of Inca social and political hegemony across newly incorporated territories. Given the provenience of the aBM and its age, it is possible the samples were local or alternatively were introduced into western highland Bolivia from the Inca core area – modern Peru. To test this hypothesis, we first evaluated the genetic relatedness of archaeological and contemporary modern maize samples using a genome sequence approach to assist in identifying the origin(s) of aBM and the relatedness to other maize from different locations. Our f3-statistics results with *Tripsacum* as outgroup suggests that aBM is closely related to two archaeological lowland Peruvian maize (Z65 and Z61) among all other ancient maize and modern maize (*Figure 4*). These results indicate that the closest relative of aBM is the archaeological maize that came from Peru. In addition, to predict the geographic location of aBM, we used spatial auto-correlations in genetic data, which allowed us to compare it to a set of archaeological maize samples of known geographic origin (*Battey et al., 2020*). Our results indicate the potential geographic location that aBM may have originally come from (locator.py –seed 549657840) and the predicted location is inside of Peru and very close to archaeological maize Z61 (*Figure 5*; R2(x)=1.0, R2(y)=1.0). However, this did not indicate that aBM originally came from the Inca core area, given that the results are based on predictions and the ages of archaeological samples vary. This suggests that aBM might not only be genetically, but also potentially geographically related to the archaeological maize from ancient Peru. This is interesting, as historically, descriptions of *capacocha* rituals showed that they were able to source their food from the stations connecting high-altitude ritual sites and other regions of the empire (*Wilson et al., 2007*). Indeed, data generated herein show that our aBM has a strong relationship with archaeological Peruvian maize in terms of both genetic identity and geographic location (*Kistler et al., 2018*). This finding is also consistent with the above previous research revealing that one of the maize groups from Peru during Inca peak times spread into newly conquered territories, then slowly diverged to fit the local environment *Vallebueno-Estrada et al., 2023* following Inca cultural influence (*Doutriaux, 2001*). After this, diverged groups were introduced outside of Inca-conquered territories. Although the origin of the aBM – either Inca core area or local – is uncertain, as our results cannot determine the dispersal direction, they do indicate that aBM has a strong relationship with archaeological Peruvian maize.

To investigate the genomic data from the aBM in relation to other archaeological and modern maize, principal component analysis (PCA) was used. As expected, the results showed that all the archaeological maize clustered with our aBM in the same cluster (*Figure 6*). SM3, 5, and 10 were excluded due to their limited data to meet requirements for the PCA analysis. Unexpectedly, there is one modern maize from Brazil in the ancient maize cluster that has overlap with archaeological Brazilian maize Z6, which indicates that certain genetic traits from ancient Brazilian maize might have been preserved in modern Brazilian maize populations. There is also one modern tropical maize very close to archaeological Peruvian maize Z65. Considering the geographic origin of Z65 and its lowland characters, it is not surprising that modern tropical maize shares some level of genome sequence similarity. Following the establishment of an important maize culture center in Peru, maize was domesticated and diverged into different groups, which then spread along with Inca culture (*Cook, 1925*). We found that the Central American cluster has overlap with both North and South America, which supports the distribution route of maize (*Swarts et al., 2017*; *Wang et al., 2021*; *Dillehay et al., 2008*). This finding is also consistent with previous research revealing that the original maize came from teosinte in Mexico, then dispersed northward into North America and southward into South America (*Doebley, 2004*; *Matsuoka et al., 2002*; *Lombardo et al., 2020*; *van Heerwaarden et al.,*

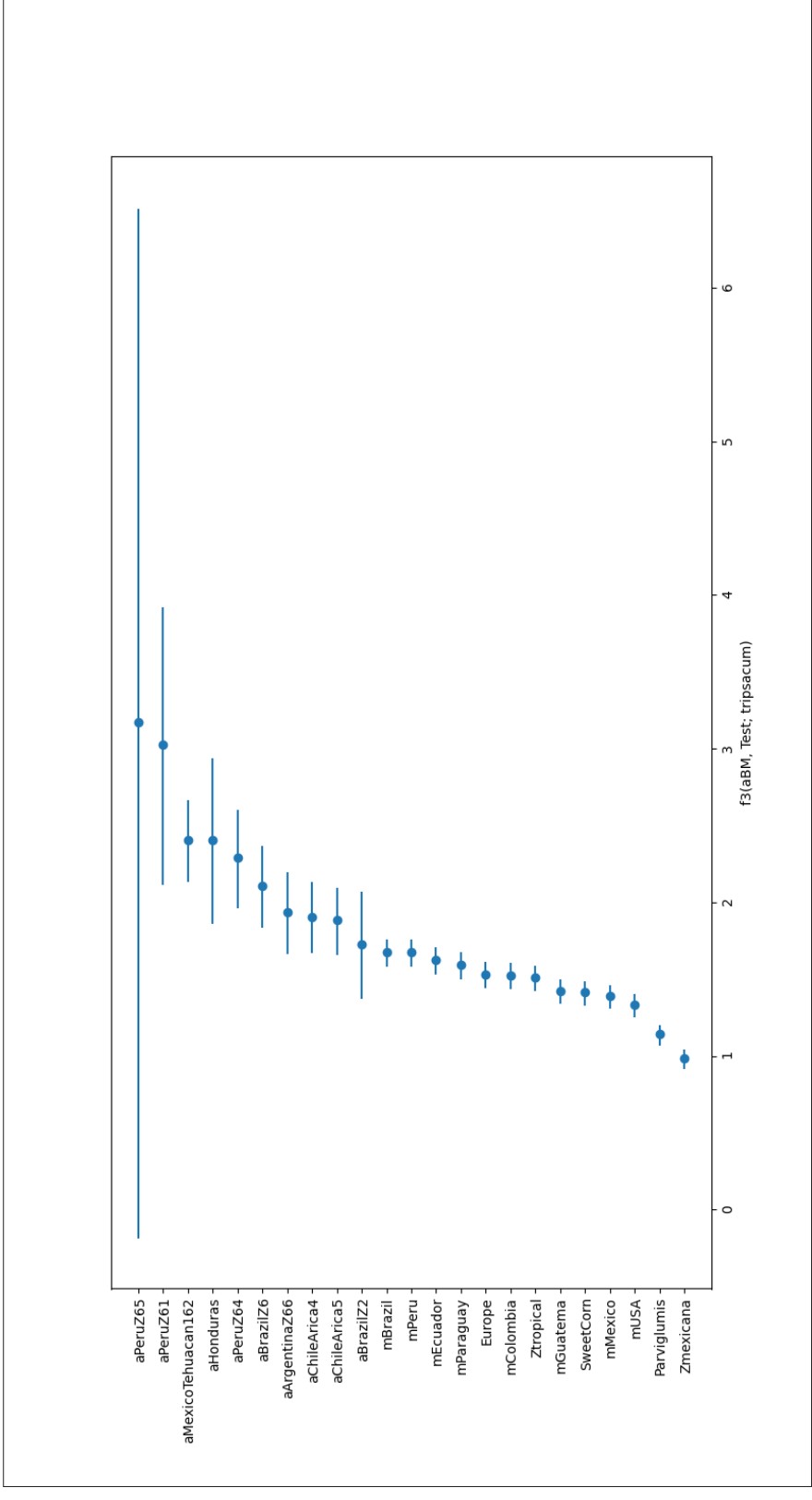

**Figure 4.** Admixture f3-statistics. In the form f3 (archaeological Bolivian maize, aBM; Test, tripsacum), where Test represents archaeological and modern samples.

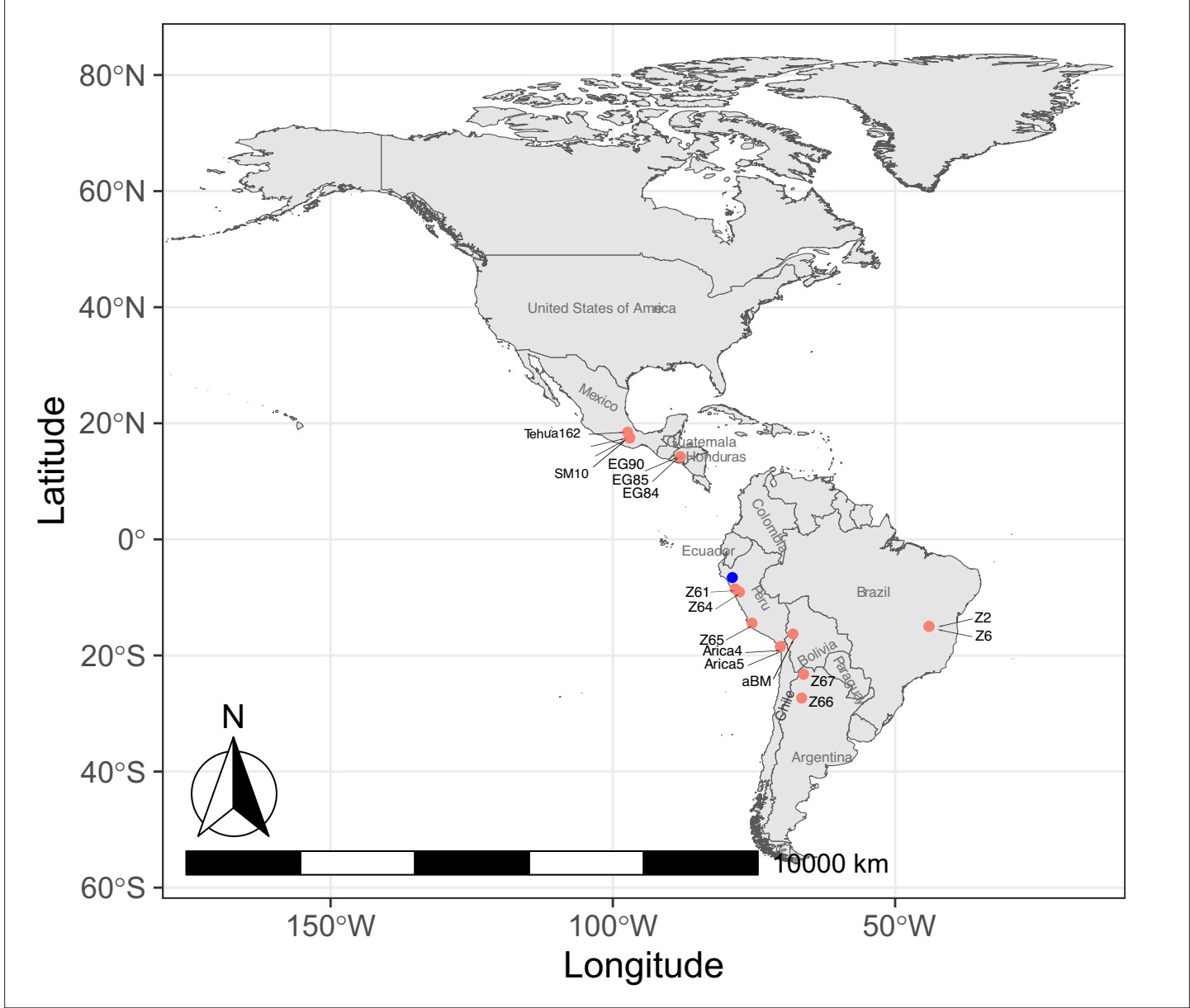

**Figure 5.** Predicted geographic location of archaeological Bolivian maize (aBM). The blue point is the predicted locations of aBM based on the single-nucleotide polymorphisms (SNPs) from all archaeological maize. The pink points are the archaeological samples.

*2011*; *Mercer et al., 2008*). We also found that a few types of maize from Europe have a much closer distance to the archaeological maize cluster compared to other modern maize, which indicates maize from Europe might be expectedly share certain traits or evolutionary characteristics with ancient maize. It is also consistent with the historical fact that maize spread to Europe from multiple areas and time periods after Christopher Columbus's late 15[th] century voyages to the Americas (*Rebourg et al., 2003*). But as shown, maize also has diversity inside the European maize cluster. It is possible that European farmers and merchants may have favored different phenotypic traits, and the subsequent spread of specific varieties followed the new global geopolitical maps of the Colonial era (*Haberer et al., 2020*; *Tenaillon and Charcosset, 2011*).

Taken together, these results support the possibility that aBM was introduced into western highland Bolivia from the Inca core area – modern Peru, and aBM may possibly reveal a history of adaptation to tropical conditions through local agency and/or following cultural influence of the Inca empire (*Mercer et al., 2008*; *Cerda-Hurtado et al., 2018*). After this, maize, both as a primary food and as

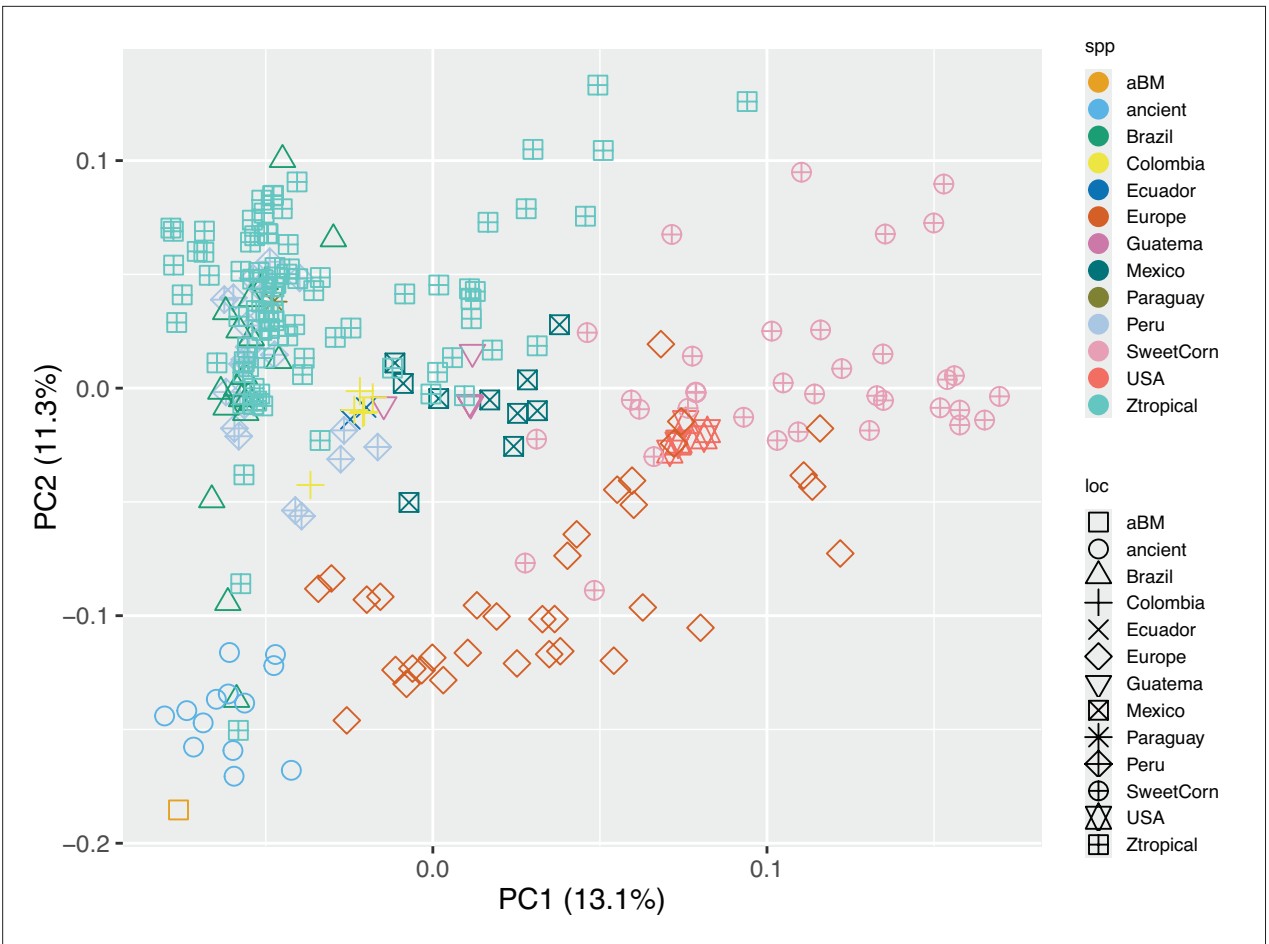

**Figure 6.** Principal component analysis (PCA) of nuclear genomic data from archaeological and modern maize. Samples with greater affinity to one another cluster closer together. Samples with the same color and shape indicate they came from the same country.

The online version of this article includes the following source data for figure 6:

**Source data 1.** Modern sample information.

a symbol of power, was dispersed from Peru throughout the Inca empire, in the end enhancing maize diversity in South America.

## Selection for aBM under the Inca culture influence

After establishing the identity of aBM and its relatedness to other regional samples, we explored how aBM was possibly shaped by human behavior, and specifically how cultural selection within the Inca empire influenced maize diversity in South America. To address this question, we examined the evolutionary relationship between aBM, its wild ancestor, and extant landraces. We constructed a Neighbor-Joining (NJ) and maximum-likelihood (ML) phylogenetic trees using the common genome-wide polymorphisms shared among aBM and modern maize samples (*Grzybowski et al., 2023*). Our data included 226 samples of modern maize, 88 modern samples of Teosinte (*Zea mays* ssp. *parviglumis*), 79 modern samples of Mexicana teosinte (*Zea mays* ssp. *mexicana*), and 1 *Tripsacum dactyloides* as an outgroup, totaling 28,588 genome-wide SNPs. As shown in *Figure 7a and b*, both phylogenetic trees indicate that aBM lies outside the diversity of modern maize yet clusters within a specific clade alongside one modern tropical maize. This suggests that aBM shares a closer genetic relationship with modern tropical maize while being genetically distinct from other modern maize. Considering modern tropical maize is adapted to tropical climates exhibiting traits such as drought tolerance, heat resistance, and the ability to thrive in low-fertility soils, it is plausible that aBM might share these stress response characteristics with tropical maize (*Chakradhar et al., 2017*; *Menkir et al., 2024*).

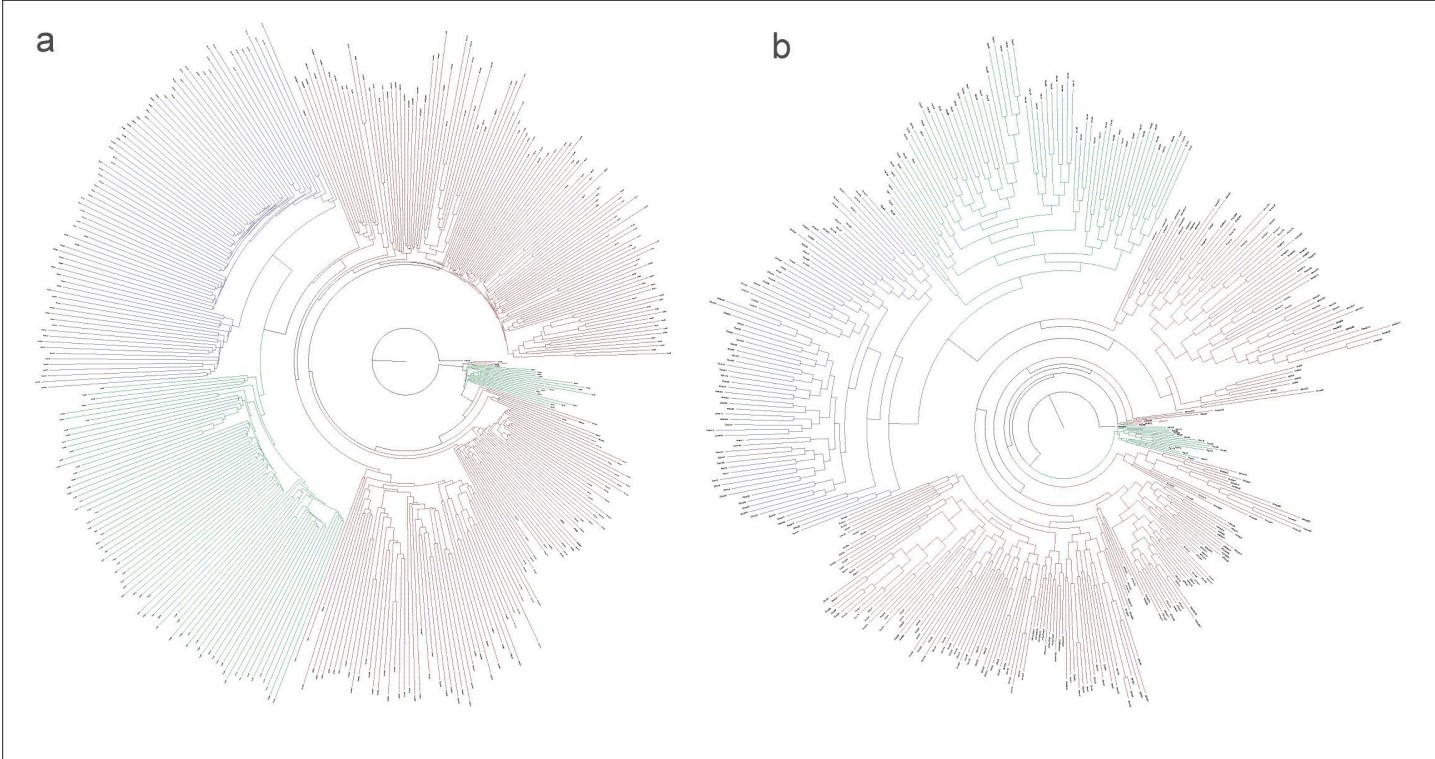

**Figure 7.** Phylogenetic trees of archaeological Bolivian maize (aBM). (**a**) Neighbor-Joining tree with VCF2Pop. (**b**) Maximum likelihood (ML) tree with SNPhylo. Parv (*Zea mays* ssp. *parviglumis*), Zmex (*Zea mays* ssp. *mexicana*), Europe (modern Europe), SweetC (model sweet corn), Ztrol (modern Tropical maize), mMexico (modern Mexico), mParaguay (modern Paraguay), mBrazil (modern Brazil), mPeru (modern Peru), mEcuador (modern Ecuador), mGuatemala (modern Guatemala), and mColombia (modern Colombia). Black color is Tripsacum. Red color is the aBM. Red-brown color is modern maize (*Zea mays* ssp. *mays* L.). Green color is Teosinte (*Zea mays* ssp. *parviglumis*). Blue color is Mexicana teosinte (*Zea mays* ssp. *mexicana*).

To investigate the selection preference on aBM under Inca cultural influence, we compared the allele frequency between aBM and other ancient maize samples. We identified 18,668 SNPs unique to aBM. Notably, two genes - Zm00001eb123120 and Zm00001eb178600-associated with traits 'internode length below ear' and 'nodes above ear' traits (*Wallace et al., 2014*) contain several unique SNPs in aBM (Dataset S2). This suggested that height-related traits in aBM possibly may have been selected under Inca cultural practices. Additionally, variations in the 5 or 3 untranslated regions UTR can significantly affect gene expression. We found 83 target genes harboring unique aBM SNPs in their UTR regions (Dataset S3). Gene Ontology (GO) results showed that three of these genes are involved in glycosyltransferase activity, while several of others are involved in defense response to biotic and abiotic stress and starch biosynthetic processes (*Source data 3*). Previous research in maize evolution implied that multiple loci are involved in shaping traits variation (*Doebley and Stec, 1991*; *Lukens and Doebley, 1999*). Combining all this information, we inferred those traits associated with plant development and stress response processes might have been selected in maize during the period of Inca cultural influence, although the selection direction is uncertain.

Finally, to understand positive selection in modern maize compared to ancient maize, we investigated the selective sweep using cross-population extended haplotype homozygosity (XP-EHH) method (*Figure 8*). Following the same method of previous research (*Brodie et al., 2016*), the cutoff metric for significant signatures of selection was set at a value of 6 for 5 kb windows across the genome. We only identified one target gene (*Figure 9*): Zm00001eb105060, whose biological process functions include plant ovule development, chromosome segregation, and asparaginyl-tRNA aminoacylation. Ovule development plays a crucial role in improving maize breeding strategies via higher seed production and hybrid vigor (*Yang and Tucker, 2021*). The positive selection of this gene in modern maize suggests that characteristics aiming for higher seed production were selected during maize diversification after aBM. This finding might relate to breeding efforts to improve seed yield in modern maize compared to ancient varieties.

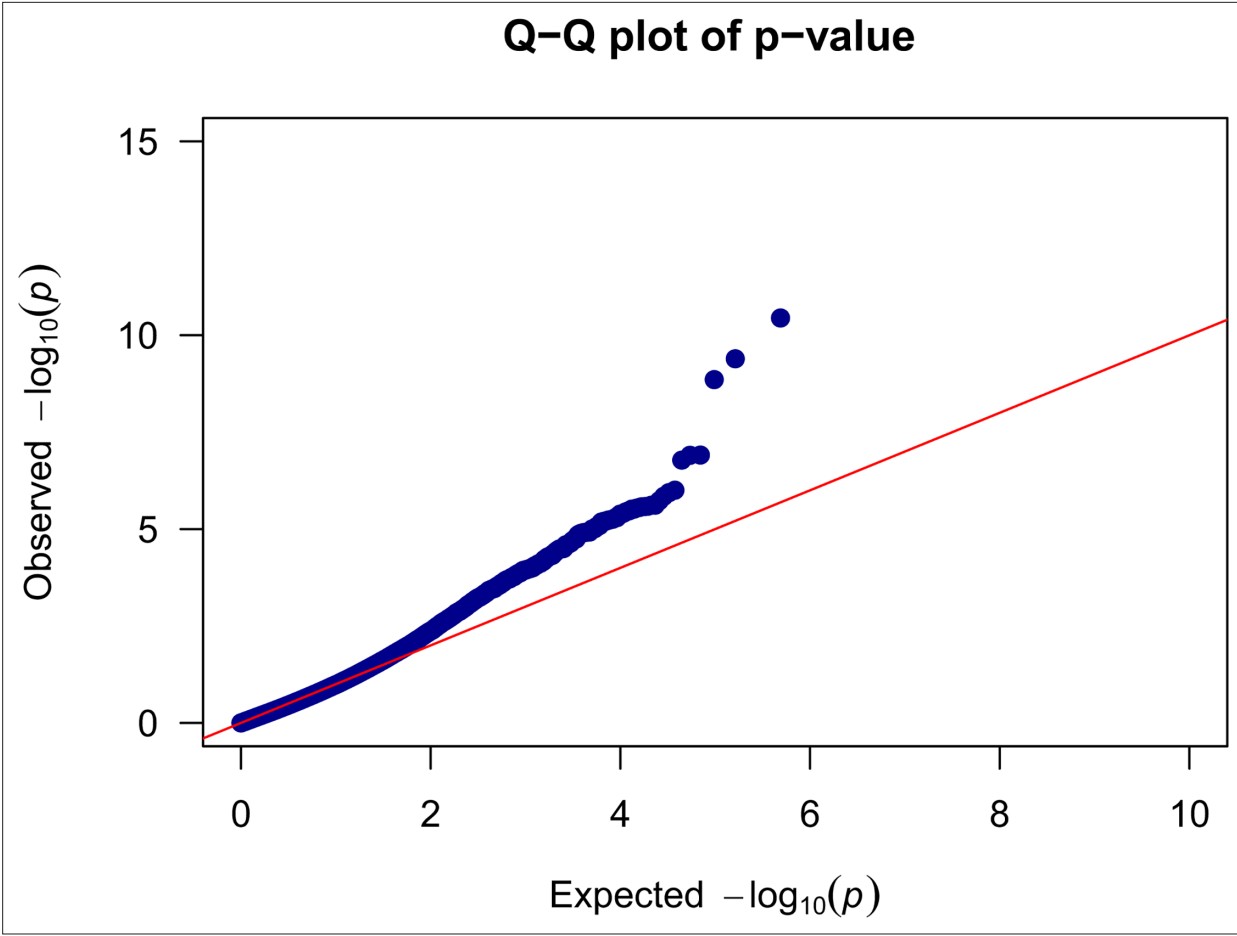

**Figure 8.** Q-Q plot for genome-wide selection in modern maize compared to archaeological maize. The Q-Q plot between modern and archaeological maize. Each point indicates a value from expected and observed. Red line represents the expected distribution of the p-value, while the blue trend represents the observed distribution.

In conclusion, the aBM included in the funerary accoutrements of a young elite Andean girl belongs to the family of ancient Peruvian maize groups, and likely was introduced to the Aymara region following Inca expansion in the late 15th century. Following its arrival in the area of western Bolivia, the Peruvian maize variety was further modified by the local population. Some genomic changes reflect selection to mitigate local environmental conditions, including response to stress. Thus, this case study supports the idea that patterns of regional diversity in pre-Contact South American maize reflect complex biocultural processes. These data may also inform subsequent studies about Colonial dispersals and the origin of global maize varieties (*Rebourg et al., 2003*).

## Discussion

Previous archaeological and anthropological studies have demonstrated that cultural and political systems shaped access to and preferences for food. In our study, we explored the evidence for the contribution of Inca culture to maize diversity in South America by analyzing 16 archaeological maize samples spanning at least 5000 years of evolution, and 226 modern maize samples. Recognizing their shared central Andean origins, we propose that the aBM sample potentially came from ancient Peruvian archaeological maize. While our findings are consistent with both genetic and archaeological evidence, they raise several questions. Notably, what specific traits were selected in aBM, and how this selection occurred under Inca cultural practices is unclear, especially given our limited aBM sample size and the lack of the same genotype in archaeological maize under non-Inca culture influence.

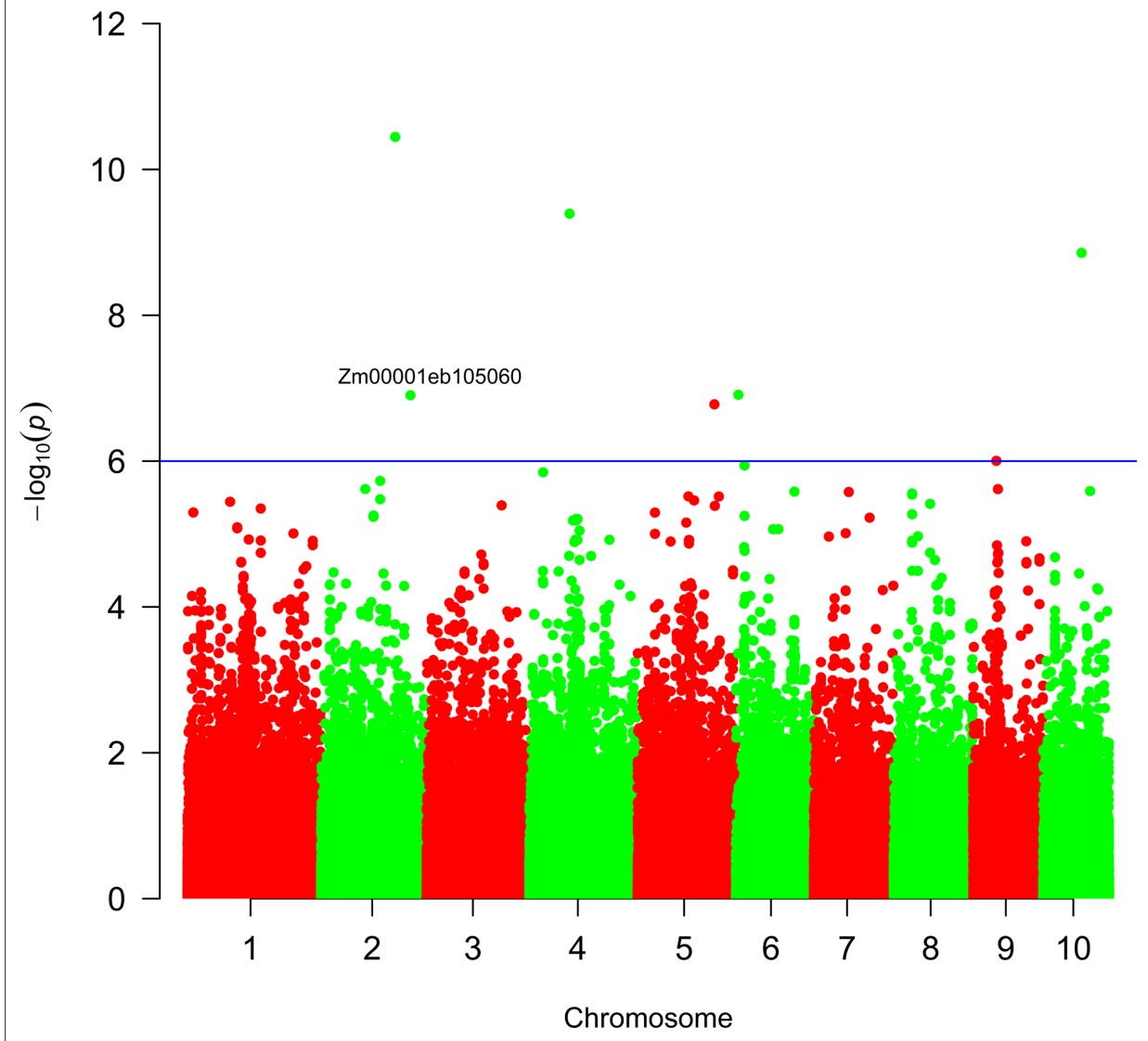

**Figure 9.** Genome-wide distribution of ancient maize-specific selective sweeps. A signal of selection of modern maize compared to archaeological maize population. Each point indicates a single-nucleotide polymorphism (SNP). The blue horizontal line shows the genome-wide significance level ($p=1\times10^{-6}$).

Cultural preferences are often a factor in maintaining certain aesthetically desirable traits (**Bellon, 1996**; **Knapp and Zimmerer, 1997**), such as color and texture, even at the cost of reduced cultivability or nutrition (**Louette and Smale, 2000**). This suggests it is possible that culture played an important role in maize evolution and diversity. To better understand how exactly cultural interactions influenced maize diversity in the Central Andes, we explored the possible origin of aBM and its closest genetic relationships to ancient maize. We identified that aBM potentially have originated from ancient Peru, and archaeological maize from Peru has the closest genetic relationship with aBM, indicating that aBM may be a product of Inca cultural dissemination into new territories (**Doutriaux, 2001**).

We further investigated which traits were possibly selected under the Inca culture by analyzing unique SNPs in aBM. The results show that traits related to plant development, and stress response were selected. These traits can aid in resisting drought and frost in the relatively harsh conditions of the Andean highlands. The selected traits could facilitate fermentation for the creation of *chicha* to meet the consumption requirement of elite lifeways (**Bray, 2003**). However, we are uncertain if these traits were directly selected under Inca culture due to the limited aBM sample size, which prevented us from identifying positive selection directly between aBM and other archaeological maize groups.

Traditional breeding systems have increased tolerance to biotic and abiotic stress and contributed to yield in crops (*Galluzzi et al., 2020*; *Kim and Lee, 2023*). In this study, we compared modern maize to archaeological maize across the entire Western Hemisphere and found that the ovule development traits were selected for breeding to improve maize seed production during its evolution. However, it is uncertain whether this selection happened directly in modern maize when compared to ancient maize, or if it happened during the process of modern improvement.

Our results not only highlight the influence of culture on maize diversity in South America but also point to the adaptive characteristics of maize in improving maize seed production during evolution. This provides a potential source for maize diversity studies and breeding programs. Most importantly, this work suggests that cultural influence plays an important role in aiding maize diversity, alongside the traits selected by early farmers.

## Materials and methods

### Plant material

The two aBM kernels were collected prior to 1890 AD from the area south of modern La Paz, Bolivia, and later curated at the Michigan State University Museum until their repatriation in 2022 (*Lovis, 2019*). AMS assay dates reveal a mid-late 15$^{th}$ century date for the maize samples. In addition, data from 16 archaeological samples collected from previously published research were also used, including two from Chile dating from 750 to 1100 BP, two from Brazil dating from 510 to 690 BP, three from Peru dating from 600 to 1000 BP, two from Argentina, dating from 980 to 1040 BP and from 70 to 130 BP, three from Honduras dating from 1740–2300 BP, and three from Mexico dating to 5310 BP (*Supplementary file 1, table S1*; *Kistler et al., 2018*; *Kistler et al., 2020*).

### aDNA extraction

Archaeological maize kernels were processed in a clean laboratory environment at Michigan State University. Whole kernels were crushed in a mortar and pestle, to which 600 μL of fresh PTB (*N*-phenacylthiazolium bromide) buffer (0.4 mg/mL proteinase K, T2.5 mM PTB, 50 nM dithiothreitol) was added. The sample was then added to 1260 μL of lysis buffer. An additional 600 μL of PTB buffer was used to rinse the mortar and added to the lysis buffer. The individually ground maize kernels were rotated overnight at 37 °C. The remaining DNA extraction followed *Swarts et al., 2017*.

### Library construction, amplification, and sequence

DNA extracts were converted to Illumina-compatible paired-end DNA libraries in a clean setting using SMARTer ThurPLEX DNA-seq from Takara Bio and amplified with seven polymerase chain reactions (PCR) cycles. The generated libraries were cleaned to remove remaining adaptors with AxyPrep Mag beads (Fisher Scientific) and analyzed on a 2100 Bioanalyzer (Agilent Technologies) to check for quality. The libraries were indexed to facilitate pooling on the NovaSeq SP at the University of Illinois (Urbana-Champaign).

### Geographic location map

The geographical coordinates of each sample were obtained from previously published studies, as indicated (*Supplementary file 1*). Utilizing latitude and longitude information, the samples were plotted on a geographic map, resulting in the successful mapping of 17 ancient samples. Geographic data visualization was carried out in R using the ggplot2, sf, leaflet, rnaturalearth, rnaturalearthdata, ggspatial, and rnaturalearthhires packages.

### Genome analysis

For the aBM sequence samples data with six libraries, we combined them into a single composite sample, referred to as aBMComb. This was achieved by mapping each individual aBM sequence sample to the B73 V5 soft-masked maize genome and subsequently removing duplicate reads before merging them. We merged the sequence sample to obtain higher coverage for the analysis that was needed for this study.

Subsequently, we used MapDamage v.2.2.1 (*Jónsson et al., 2013*) verified cytosine deamination profiles consistent with ancient DNA, and estimated the misincorporation frequencies. Based on these

results, we hard-masked 5′ thymine and 3′ adenine residues within 5 nt of the two ends, where deaminations was most concentrated.

Data for 16 ancient samples with specific geographic location information (maize landraces) obtained from previously published work (*Kistler et al., 2018*; *Kistler et al., 2020*; *Vallebueno-Estrada et al., 2016*) by downloading sequencing from Sequence Read Archive (SRA) (https://www.ncbi.nlm.nih.gov/sra/) (*Chia et al., 2012*). To ensure consistency and accuracy, we downloaded the raw sequence (SRA format) for each sample and converted it into FASTQ format using fastq-dump.

For the archaeological maize samples, adapters were removed and paired reads were merged using AdapterRemoval (*Lindgreen, 2012*) with parameters `--minquality 20 --minlength 30`. All 5′ thymine and 3′ adenine residues within 5 nt of the two ends were hard-masked, where deamination was most concentrated. Reads were then mapped to soft-masked B73 v5 reference genome using BWA v0.7.17 with disabled seed (-l 1024 -o 0 -E 3) and a quality control threshold (-q 20) based on the recommended parameter (*Schubert et al., 2012*) to improve ancient DNA mapping. Duplicated reads were removed by GATK tools Picard (https://gatk.broadinstitute.org/hc/en-us). The coverage of each nucleotide position was estimated by bam-readcount (https://github.com/genome/bam-readcount copy archived at *Ferrero, 2022*; *Supplementary file 1,table S2*), and the overall coverage of the assembled genome sequence was measured using Qualimap v2.2.1 (*Okonechnikov et al., 2016*).

## Variant calling

We used SequenceTools pileupCaller (v 1.5.4.0; https://github.com/stschiff/sequenceTools; *Schiffels, 2025*) to generate pseudohaploid calls for 17 archaeological maize samples. The commands executed were: samtools mpileup -R -B -q20 -Q20 & pileupCaller `--sampleNames` <ancient sample name >randomHaploid `--singleStrandMode`. This process yielded 16,815,782 SNPs.

We further filtered the SNPs using estimated information from bam-readcount. SNPs supported by at least two reads were retained for each individual archaeological maize sample. We then combined the filtered SNPs from each sample, keeping only those SNPs that were present in at least one sample. This resulted in a final dataset of 2,808,917 SNPs for the 17 archaeological maize samples.

For all subsequent analyses, the ancient genotype data was merged with the genotype dataset for 394 samples from Grzybowski et al. ('Feb 03, 2023 version files' in Dryad repository: https://doi.org/10.5061/dryad.bnzs7h4f1). The merged dataset included 2,635,047 SNPs from chromosomes 1–10 that were common in both archaeological and modern maize samples, which were used for downstream analysis.

## *f3* statistic and phylogenetic analyses

We created a genotype file by combining 17 archaeological maize and publicly available SNP dataset from Grzybowski et al, resulting in a total of 17-archaeological and 226-modern maize, 88-modern samples of Teosinte (*Zea mays* ssp. *parviglumis*), 79-modern samples of *Mexicana teosinte* (*Zea mays* ssp. *mexicana*), and 1-*Tripsacum* genotypes.

Next, we used plink v1.9 (*Purcell et al., 2007*) to prune for linkage disequilibrium with the parameters `--indep-pairwise 50 5 0.1`, following previous research (*van Heerwaarden et al., 2011*), to prepare the dataset for PCA using 'plink -pca.' The final results were plotted in R using the ggplot2 package (*Wickham, 2009*).

We calculated the *f3* statistic (*Green et al., 2010*; *Durand et al., 2011*) using the admixr package (*Petr et al., 2019*), which is part of the package of ADMIXTURE software suite (*Alexander and Lange, 2011*). The outgroup f3 statistics was used to measure how closely aBM and each test group are related to each other. The final results were plotted using Python.

A total of 28,588 SNPs present in both aBM and all modern maize, out of a total of 2,626,087 SNPs, was used to build the phylogenetic tree. A maximum likelihood (ML) phylogenetic tree was constructed using SNPhylo (*Lee et al., 2014*) with parameters: -v snps.vcf -r -o tripsacum -P tree -b 1000; A Neighbour-Joining (NJ) phylogenetic tree was constructed using VCF2PopTree (*Subramanian et al., 2019*) with the 'number of differences' model. Both ML and NJ phylogenetic trees were inferred and visualized using FigTree v1.4.4 (http://tree.bio.ed.ac.uk/software/figtree/).

## Geographical location prediction

To predict the possible geographical location of aBM, we applied a deep neural network method (*Battey et al., 2020*) using only 17 ancient maize samples, as all the archaeological maize clustered in the same cluster. A total of 2,800,190 SNPs were used to predict the possible geographical location of aBM. The command used was: locator.py `--vcf SNP_file --sample_data sample_data.txt -out name`.

## Infer selection on aBM

Due to the lack of archaeological maize with the same genotype as aBM that grows outside the influence of Inca culture, it was impossible to directly identify the selection that occurred on aBM under Inca cultural influence. To determine if any of our SNPs are related to known traits, we utilized existing information on SNP-trait associations (*Wallace et al., 2014*). To achieve potential unique selection on aBM, we calculated the allele frequency for each SNPs between aBM and other ancient maize, resulting in allele frequency data for 49,896 SNPs. Of these, 18,668 SNPs were unique to aBM. We consulted SNPVersity 2.0 from MaizeGDB (*Grzybowski et al., 2023*; *Schnable et al., 2009*) and found that 84 of these unique SNPs were classified as '3_prime_UTR_variant' or '5_prime_UTR_variant.' These SNPs corresponded to 83 target genes, for which we retrieved Gene Ontology (GO) terms.

## Identification of selection sweeps

To detect genomic signatures of recent selective sweeps in modern maize compared to ancient maize groups, SNPs that are present in at least two archaeological samples and modern samples were used. We applied the cross-population extended haplotype homozygosity (XP-EHH) (*Sabeti et al., 2007*) method. A commonly used threshold $-\log_{10}(p)=6$ was set as the cutoff threshold for identifying the signal of significant selection. The regions with $-\log10(p) \geq 6$ were designated as candidate sweeps for candidate genes located within 5 KB of these regions identified, following the approach used in previous research (*Brodie et al., 2016*). MaizeMine from MaizeGDB (*Lawrence et al., 2004*) was used to obtain putative gene function information.

# Acknowledgements

We are grateful to all participants who were involved in this project. We are thankful for the support of the Michigan State University Cloud Fellowship for computational skills training. This work was supported in part through computational resources and services provided by the Institute for Cyber-Enabled Research at Michigan State University. We thank Jennifer Wai, University of Chicago, for her help with DNA extraction and library preparation. We also thank Emily Josephs, Michigan State University, and Logan Kistler, Smithsonian Institution of the National Museum of Natural History, for their help with the DNA analysis. Thanks to José Capriles Flores for his insightful reading and comments. The MSU Museum provided maize samples and images for documentation and analysis prior to repatriation to Bolivia. Research in the laboratory of Brad Day was supported by the MSU Research Foundation.

# Additional information

### Funding

| Funder | Grant reference number | Author |
|---|---|---|
| Michigan State University Foundation | Michigan State University Foundation Professor Endowment | Huan Chen Brad Day |

The funders had no role in study design, data collection and interpretation, or the decision to submit the work for publication.

## Author contributions
Huan Chen, Data curation, Formal analysis, Visualization, Methodology, Writing – original draft, Writing – review and editing; Amy Baetsen-Young, Conceptualization, Formal analysis; Addie Thompson, Thelma Madzima, Sally Wasef, Claudia Rivera Casanovas, Methodology; Brad Day, Conceptualization, Project administration; William Lovis, Conceptualization, Investigation, Writing – original draft, Writing – review and editing; Gabriel Wrobel, Conceptualization, Investigation, Writing – original draft, Project administration, Writing – review and editing

## Author ORCIDs
Huan Chen (ORCID) https://orcid.org/0000-0002-6584-8489
Addie Thompson (ORCID) https://orcid.org/0000-0002-4442-6578
Brad Day (ORCID) https://orcid.org/0000-0002-9880-4319
Thelma Madzima (ORCID) https://orcid.org/0000-0002-7114-8454
Sally Wasef (ORCID) https://orcid.org/0000-0002-7207-7395
William Lovis (ORCID) https://orcid.org/0000-0002-9221-7447
Gabriel Wrobel (ORCID) https://orcid.org/0000-0002-1821-3202

Reviewer #1 (Public review): https://doi.org/10.7554/eLife.106818.3.sa1
Reviewer #2 (Public review): https://doi.org/10.7554/eLife.106818.3.sa2
Author response https://doi.org/10.7554/eLife.106818.3.sa3

# Additional files

## Supplementary files
Supplementary file 1. (A) Geographical information of the archaeological maize samples used in this study. (B) Paleogenomic characterization of archaeological Bolivian maize sequence samples with six libraries. (C) The percentage of genomic sites covered at variable depths in the archaeological Bolivian maize (aBM) sample 766 (aBMComb_rm5nt.rmDR.bam).

Source data 1. Known single-nucleotide polymorphisms (SNPs) related to traits.

Source data 2. Single-nucleotide polymorphisms (SNPs) information from SNPVersity.

Source data 3. Gene Ontology (GO) term enrichment information.

MDAR checklist

## Data availability
The authors declare that all data supporting the findings of this study are included in the manuscript and its supplementary files. Source data are provided with this paper. The genome sequence data of ancient Bolivian Maize has been deposited in the NCBI database under BioProject accession PRJNA886637. Custom scripts and single nucleotide polymorphism (SNP) calls are available on Dryad (DOI: https://doi.org/10.5061/dryad.w6m905qtd).

The following datasets were generated:

| Author(s) | Year | Dataset title | Dataset URL | Database and Identifier |
| --- | --- | --- | --- | --- |
| Chen H, Baetsen-Young A, Thompson A, Brad D, Lovis W, Wrobel G, Madzima T, Wasef S, Rivera Casanovas C | 2025 | 15th century CE Bolivian maize reveals genetic affinities with ancient Peruvian maize | https://doi.org/10.5061/dryad.w6m905qtd | Dryad Digital Repository, 10.5061/dryad.w6m905qtd |
| Chen H, Baetsen-Young A, Day B, Lovis W, Wrobel G | 2022 | Zea mays | https://www.ncbi.nlm.nih.gov/bioproject/?term=PRJNA886637 | NCBI BioProject, PRJNA886637 |

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
