## [Editor Report · eLife Assessment]

This **useful** study attempts to place an ancient maize sample from Bolivia, dated to the end of the Incan empire, in genetic and geographical context. The analyses show that this sample is most closely related to ancient Peruvian maize, but the data remain **inadequate** to determine the direction of dispersal and the extent of Inca influence over the genetic make up of the analyzed sample. There are additional deficiencies in the statistical analyses and selection inferences. The topic of the study would appeal to researchers studying maize dispersal and adaptation.

---

## [Referee Report · Reviewer #1 (Public review)]

Summary:

In this manuscript, authors describe a good quality ancient maize genome from 15th century Boliva and try to link the genome characteristics to Inca influence. Overall, the revised manuscript is still below the standard in the field. While dating of the sample and the authentication of ancient DNA has been evidenced robustly, the downstream genetic analyses do not support the conclusion that genomic changes can be attributed to Inca influence. There is more story telling than story testing in this manuscript, analyses are not robust and possibly of very narrow interest.

Strengths:

Technical data related to the maize sample are robust. Radiocarbon dating strongly evidenced sample age, estimated to around 1474 AD. Authentication of ancient DNA has been done robustly. Spontaneous C-to-T substations which are present in all ancient DNA are visible in reported sample with the expected pattern. Despite low fraction of C-to-T at the 1st base, this number could be consistent with cool and dry climate in which the sample was preserved. The distribution of DNA fragment sizes is consistent with expectations for sample of this age.

Weaknesses:

(1) The geographic placement of the sample based on genetic data is not robust. To make use of the method correctly, it would be necessary to validate that genetic samples in this region follow the assumption of the 'isolation-by-distance' with dense sampling, which has not been done. Without this important information, we do not know if genetic similarity is influenced by demographic events and/or selection. The analysis is not a robust evidence of sample connectivity.

(2) The conclusion that Ancient Andean maize is genetically similar to European varieties and hence share similar evolutionary history is not well supported. PCA plot in Fig. 4 merely represents sample similarity based on two components (jointly responsible for about 20% of variation explained). Contrary to authors' conclusion, the direct test of similarity using outgroup f3 statistic does not support that European varieties are particularly closely related to ancient Andean maize. These levels of shared drift could be due ancient Andean maize relationship with other related groups, such as ancient or modern Brazil. A relationship test between multiple populations would be necessary to show significant direct relationship between ancient Andean maize and European maize.

(3) The conclusion that selection detected in aBM sample is due to Inca influence has no support. Firstly, selection signature can be due to environmental or any other factors. To disentangle those, authors would need to generate the data for a large number of samples from similar cultural context and from a wide-ranging environmental context followed by a formal statistical test. Secondly, allele frequency increase can be attributed to selection or demographic processes, and alone is not a sufficient evidence for selection. Presented XP-EHH method seems unsuitable for single individual. Overall, methods used in this paper raise some concerns: (i) how accurate are allele-frequency tests of selection when only single individual is used as a proxy for a whole population, (ii) the significance threshold has been arbitrary fixed to an absolute number based on other studies, but the standard is to use, for example, top fifth percentile.

In sum, this manuscript presents new data that seem to be of high quality, but the analyses are frequently inappropriate and/or over-interpreted.

---

## [Referee Report · Reviewer #2 (Public review)]

I am glad to see a revised version of the manuscript. The authors have successfully handled some of my comments, but others require additional attention. In particular, the dataset seems quite robust and valuable to publish, and the descriptive analysis of its position relative to other modern and ancient genomes is generally sound. The selection analyses remain unsupported, and should be removed from the paper. In addition, I agree with the other reviewers and reiterate my comment that the Locator analysis is not robust.

As I said in my original review, the XP-EHH method is not applicable to pseudohaploid variant calls in a single individual. This method is simply not appropriate to apply to the data at hand, as the method relies on knowledge of diploid genotypes, usually phased, and the results from this test are not robust. It is possible that the XP-EHH method could be extended to this data type or genotype likelihoods with extensive validation and conditioning on a large reference panel, but in general haplotype-based approaches have not been extensible to low-coverage pseudohaplotype datasets. At any rate, any off-the-shelf implementation is inappropriate and unsupported. I am sorry to be this negative about this analysis, but it cannot be used as presented, the results from using it in this way would be spurious by definition.

In addition, identifying GO terms without statistical assessment of enrichment is not a robust analysis, nor is selecting genes with a high proportion of rare alleles without extensive additional contextualization based on the expectations of neutrality and deviations potentially tied to selection. For this reason, the two genes linked with height traits have no support here as genuinely being targets of selection. It is a frustrating reality for us in the ancient DNA field that small numbers of highly degraded genomes offer extremely limited scope for selection analyses, but that's the unfortunate state of play, and is the situation here.

My other major critique remains the application of the Locator method. As Reviewer 1 notes, this method must be built on a densely sampled dataset with strong isolation by distance, which is not done here. The authors explained their approach with more detail in their response, but it is fundamentally inappropriate for this dataset. It does not add anything more than the f3 analysis, and creates a falsely precise inference of genetic-geographic origins that is not supported.

Per authors' response to my previous recommendation 6, it is not advisable to re-map the reads after damage masking, and doing this with a conservative hard-masking approach will lead to a high mismatch rate and significant loss of reads in BWA. This could also exacerbate reference sequence bias which is already a major challenge for ancient DNA (see Gunther et al 2019 PLoS Genet). The correct approach is to map reads, mask or rescale for damage, and then proceed with the modified alignment file. In response to Reviewer 3's comment 3, the authors also refer to a "0 mismatch alignment" strategy. This is not concordant with the damage analysis, and if they truly do not allow mismatches this would be very inadvisable, as it would allow an extreme reference sequence bias.

---

## [Author Response]

The following is the authors’ response to the original reviews.

**Reviewer #1 (Public review):**
Summary:In this manuscript, the authors describe a good-quality ancient maize genome from 15th-century Bolivia and try to link the genome characteristics to Inca influence. Overall, the manuscript is below the standard in the field. In particular, the geographic origin of the sample and its archaeological context is not well evidenced. While dating of the sample and the authentication of ancient DNA have been evidenced robustly, the downstream genetic analyses do not support the conclusion that genomic changes can be attributed to Inca influence. Furthermore, sections of the manuscript are written incoherently and with logical mistakes. In its current form, this paper is not robust and possibly of very narrow interest.Strengths:Technical data related to the maize sample are robust. Radiocarbon dating strongly evidenced the sample age, estimated to be around 1474 AD. Authentication of ancient DNA has been done robustly. Spontaneous C-to-T substitutions, which are present in all ancient DNA, are visible in the reported sample with the expected pattern. Despite a low fraction of C-to-T at the 1st base, this number could be consistent with the cool and dry climate in which the sample was preserved. The distribution of DNA fragment sizes is consistent with expectations for a sample of this age.Weaknesses:

Thank you for all your thoughtful comments. See below for comments on each.

(1) Archaeological context for the maize sample is weakly supported by speculation about the origin and has unreasonable claims weighing on it. Perhaps those findings would be more convincing if the authors were to present evidence that supports their conclusions: (i) a map of all known tombs near La Paz, (ii) evidence supporting the stone tomb origins of this assemblage, and (iii) evidence supporting non-Inca provenance of the tomb.

We believe we are clear about what information we have about context. First, the intake records from the MSU Museum from 1890 are not as detailed as we would like, but we cannot enhance them. The mummified girl and her accoutrements, including the maize, came from a stone tower or chullpa south of La Paz, in what is now Bolivia. We do not know which stone chullpa, so a map would be of limited use. The mortuary group is identified as Inca, but as we note the accoutrements do not appear of high status, so it is possible that she is not an elite. Mud tombs are normally attributed to the local population, and stone towers to Inca or elites. We have clarified at multiple places in the text that the maize is from the period of Inca incursion in this part of Bolivia and have modified text to reflect greater uncertainty of Inca or local origin, but that selection for environmentally favorable characteristics had taken place. Regardless, there are three 15th c CE or AD AMS ages on the maize, a cucurbita rind, and a camelid fiber. The maize is almost certainly mid to late 15th century CE.

(2) Dismissal of the admixture in the reported samples is not evidenced correctly. Population f3 statistic with an outgroup is indeed one of the most robust metrics for sample relatedness; however, it should not be used as a test of admixture. For an admixture test, the population f3 statistic should be used in the form: (i) target population, (ii) one possible parental population, (iii) another possible parental population. This is typically done iteratively with all combinations of possible parental populations. Even in such a form, the population f3 statistic is not very sensitive to admixture in cases of strong genetic drift, and instead population f4 statistic (with an outgroup) is a recommended test for admixture.

We have removed “Our admixture f3-statistics test results suggest aBM is not admixed” in our revised manuscript. Since our goal here is to identify which group(s) has(have) the highest relatedness with aBM, so population f3 statistic with an outgroup is the most robust metric to do the test and to support our conclusion here.

(3) The geographic placement of the sample based on genetic data is not robust. To make use of the method correctly, it would be necessary to validate that genetic samples in this region follow the assumption of the 'isolation-by-distance' with dense sampling, which has not been done. Additionally, the authors posit that "This suggests that aBM might not only be genetically related to the archaeological maize from ancient Peru, but also in the possible geographic location." The method used to infer the location is based on pure genetic estimation. The above conclusion is not supported by this method, and it directly contradicts the authors' suggestion that the sample comes from Bolivia.

We understood that it is necessary to validate the assumption of the 'isolation-by-distance' with dense sampling. But we did not do it because: (1) the ancient maize age ranges from ~5000BP to ~100BP and they were found in very different countries at different times. (2) isolation-by-distance is a population genetic concept and it's often used to test whether populations that are geographically farther apart are also more genetically different. Considering we only have 17 ancient samples in total our sample size is not sufficient for a big population test.

For "It directly contradicts the authors' suggestion that the sample comes from Bolivia.”, as we described in our manuscript that “Given the provenience of the aBM and its age, it is possible the samples were local or alternatively were introduced into western highland Bolivia from the Inca core area – modern Peru.” The sample recording file did show the aBM sample was found in Bolivia, but we do not know where aBM originally came from before it was found in Bolivia. To answer this question, we used locator.py to predict the potential geographic location that aBM may have originally come from, and our results showed that the predicted location is inside of modern Peru and is also very close to archaeological Peruvian maize.

Therefore, our conclusion that "This suggests that aBM might not only be genetically related to the archaeological maize from ancient Peru, but also in the possible geographic location” does not contradict that the sample was found Bolivia.

(4) The conclusion that Ancient Andean maize is genetically similar to European varieties and hence shares a similar evolutionary history is not well supported. The PCA plot in Figure 4 merely represents sample similarity based on two components (jointly responsible for about 20% of the variation explained), and European samples could be very distant based on other components. Indeed, the direct test using the outgroup f3 statistic does not support that European varieties are particularly closely related to ancient Andean maize. Perhaps these are more closely related to Brazil? We do not know, as this has not been measured.

Our conclusion is “We also found that a few types of maize from Europe have a much closer distance to the archaeological maize cluster compared to other modern maize, which indicates maize from Europe might expectedly share certain traits or evolutionary characteristics with ancient maize. It is also consistent with the historical fact that maize spread to Europe after Christopher Columbus's late 15th century voyages to the Americas. But as shown, maize also has diversity inside the European maize cluster. It is possible that European farmers and merchants may have favored different phenotypic traits, and the subsequent spread of specific varieties followed the new global geopolitical maps of the Colonial era”.

We understood your concerns that two components only explain about 20% of the variation. But as you can see from the Figure 2b in Grzybowski, M.W. et al., 2023 publication, it described that “the first principal component (PC1) of variation for genetic marker data roughly corresponded to the division between domesticated maize and maize wild relatives is only 1.3%”. It shows this is quite common in maize, especially when the datasets include landraces, hybrids, and wild relatives. For our maize dataset, we have archaeological maize data ranging from ~5,000BP to ~100BP, and we also have modern maize, which makes the genetic structure of our data more complicated. Therefore, we think our two components are currently the best explanation currently possible. We also included PCA plot based on component 1 and 3 in Fig4_PCA13.pdf. It does not show that the European samples are very distant.

For “Perhaps these are more closely related to Brazil?”, thank you for this very good question, but we apologize that we cannot answer this question from our current study because our study focuses on identifying the location where aBM originally came from, establishing and explaining patterns of genetic variability of maize, with a specific focus on maize strains that are related to our current aBM. Thus, we will not explore the story between maize from Brazil and European maize in our current study.

(5) The conclusion that long branches in the phylogenetic tree are due to selection under local adaptation has no evidence. Long branches could be the result of missing data, nucleotide misincorporations, genetic drift, or simply due to the inability of phylogenetic trees to model complex population-level relationships such as admixture or incomplete lineage sorting. Additionally, captions to Figure S3, do not explain colour-coding.

We have removed “aBM tends to have long branches compare to tropicalis maize, which can be explained by adaption for specific local environment by time.” in our revised manuscript.

We have added the color-coding information under Fig. S3 in our revised manuscript.

(6) The conclusion that selection detected in aBM sample is due to Inca influence has no support. Firstly, selection signature can be due to environmental or other factors. To disentangle those, the authors would need to generate the data for a large number of samples from similar cultural contexts and from a wide-ranging environmental context, followed by a formal statistical test. Secondly, allele frequency increase can be attributed to selection or demographic processes, and alone is not sufficient evidence for selection. The presented XP-EHH method seems more suitable. Overall, methods used in this paper raise some concerns: (i) how accurate are allele-frequency tests of selection when only single individual is used as a proxy for a whole population, (ii) the significance threshold has been arbitrary fixed to an absolute number based on other studies, but the standard is to use, for example, top fifth percentile. Finally, linking selection to particular GO terms is not strong evidence, as correlation does not imply causation, and links are unclear anyway.In sum, this manuscript presents new data that seems to be of high quality, but the analyses are frequently inappropriate and/or over-interpreted.

Regarding your suggestion that “from similar cultural contexts and from a wide-ranging environmental context, followed by a formal statistical test”, we apologize that this cannot be done in our current study because we could not find other archaeological maize samples/datasets that are from similar cultural contexts.

For “Secondly, allele frequency increase can be attributed to selection or demographic processes, and alone is not sufficient evidence for selection.” Yes, we agree, and that’s why we said it “inferred” the conclusion instead of “indicated”. Furthermore, we revised the whole manuscript following all reviewers’ comments and reorganized and reduced the part on selection on aBM.

For “The presented XP-EHH method seems more suitable”, we do not think XP-EHH is the best method that could be used here because we only have one aBM sample, but XP-EHH is more suitable for a population analysis.

For “Finally, linking selection to particular GO terms is not strong evidence, as correlation does not imply causation, and links are unclear anyway.”, as we described in our manuscript, our results “inferred” instead of “indicated” the conclusion.

**Reviewer #2 (Public review):**
Summary:The manuscript presents valuable new datasets from two ancient maize seeds that contribute to our growing understanding of the maize evolution and biodiversity landscape in pre-colonial South America. Some of the analyses are robust, but the selection elements are not supported.Strengths:The data collection is robust, and the data appear to be of sufficiently high quality to carry out some interesting analytical procedures. The central finding that aBM maize is closely related to maize from the core Inca region is well supported, although the directionality of dispersal is not supported.Weaknesses:

Thank you for your comments and suggestions. See below for responses and explanations.

The selection results are not justified, see examples in the detailed comments below.(1) The manuscript mentions cultural and natural selection (line 76), but then only gives a couple of examples of selecting for culinary/use traits. There are many examples of selection to tolerate diverse environments that could be relevant for this discussion, if desired.

We have added related examples with references supported in our revised manuscript.

(2) I would be extremely cautious about interpreting the observations of a Spanish colonizer (lines 95-99) without very significant caveats. Indigenous agriculture and food ways would have been far more nuanced than what could be captured in this context, and the genocidal activities of the Europeans would have impacted food production activities to a degree, and any contemporaneous accounts need to be understood through that lens.

We agree with the first part of this comment and have softened our use of this particular textual material such that it is far less central to interpretation.While of interest, we cannot evaluate the impact of colonial European activities or observational bias for purposes of this analysis.

(3) The f3 stats presented in Figure 2 are not set up to test any specific admixture scenarios, so it is unsupported to conclude that the aBM maize is not admixed on this basis (lines 201-202). The original f3 publication (Patterson et al, 2012) describes some scenarios where f3 characteristics associate with admixture, but in general, there are many caveats to this approach, and it's not the ideal tool for admixture testing, compared with e.g., f4 and D (abba-baba) statistics.

You make an important point that f3 stats is not the ideal tool for admixture testing. Since our study goal here is to identify which group(s) has(have) the highest relatedness with aBM, the population f3 statistic with an outgroup is the most robust metrics with which to do the test and to support our conclusion here. We have removed the “Our admixture f3-statistics test results suggest aBM is not admixed” in our revised manuscript.

(4) I'm a little bit skeptical that the Locator method adds value here, given the small training sample size and the wide geographic spread and genetic diversity of the ancient samples that include Central America. The paper describing that method (Battey et al 2020 eLife) uses much larger datasets, and while the authors do not specifically advise on sample sizes, they caution about small sample size issues. We have already seen that the ancient Peruvian maize has the most shared drift with aBM maize on the basis of the f3 stats, and the Locator analysis seems to just be reiterating that. I would advise against putting any additional weight on the Locator results as far as geographic origins, and personally I would skip this analysis in this case.

As we described in our manuscript, we have 17 archaeological samples in total. Please find more detailed information from the “geographical location prediction” section.

We cannot add more ancient samples because they are all that we could find from all previous publications. We may still want to keep this analysis because f3 stats indicates the genome similarity, but the purpose of locator.py analysis is indicating the predicted location of origin of a genetic sample by comparing it to a set of samples of known geographic origin.

(5) The overlap in PCA should not be used to confirm that aBM is authentically ancient, because with proper data handling, PCA placement should be agnostic to modern/ancient status (see lines 224-226). It is somewhat unexpected that the ancient Tehuacan maize (with a major teosinte genomic component) falls near the ancient South American maize, but this could be an artifact of sampling throughout the PCA and the lack of teosinte samples that might attract that individual.

We have removed “which supports the authenticity of aBM as archaeological maize” in our revised manuscript. The PCA was only applied for all maize samples, so we did not include any teosinte samples in the analysis.

(6) What has been established (lines 250-251) is genetic similarity to the Inca core area, not necessarily the directionality. Might aBM have been part of a cultural region supplying maize to the Inca core region, for example? Without a specific test of dispersal directionality, which I don't think is possible with the data at hand, this is somewhat speculative.

We added this and re-wrote this part in our revised manuscript.

(7) Singleton SNPs are not a typical criterion for identifying selection; this method needs some citations supporting the exact approach and validation against neutral expectations (line 278). Without Datasets S2 and S3, which are not included with this submission, it is difficult to assess this result further. However, it is very unexpected that ~18,000 out of ~49,000 SNPs would be unique to the aBM lineage. This most likely reflects some data artifact (unaccounted damage, paralogs not treated for high coverage, which are extremely prevalent in maize, etc). I'm confused about unique SNPs in this context. How can they be unique to the aBM lineage if the SNPs used overlap the Grzybowski set? The GO results do not include any details of the exact method used or a statistical assessment of the results. It is not clear if the GO terms noted are statistically enriched.

We have added references 53 and 54 in our revised manuscript, and we also uploaded the Datasets S2 and S3.

For “I'm confused about unique SNPs in this context. How can they be unique to the aBM lineage if the SNPs used overlap the Grzybowski set?”, as we described in our materials and method part that “To achieve potential unique selection on aBM, we calculated the allele frequency for each SNPs between aBM and other archaeological maize, resulting in allele frequency data for 49,896 SNPs. Of these,18,668 SNPs were unique to aBM.” Thus, the unique SNPs for aBM came from the comparison between aBM with other archaeological maize, and we did not use any modern maize data from the Grzybowski set.

For “The GO results do not include any details of the exact method used or a statistical assessment of the results. It is not clear if the GO terms noted are statistically enriched.” We did not do GO Term enrichment, so there are no statistical assessments for the results. What we have done was we retained the GO Terms information for each gene by checking their biological process from MaizeGDB, after that, we summarized the results in Dataset S4.

(8) The use of XP-EHH with pseudo haplotype variant calls is not viable (line 293). It is not clear what exact implementation of XP-EHH was used, but this method generally relies on phased or sometimes unphased diploid genotype calls to observe shared haplotypes, and some minimum population size to derive statistical power. No implementation of XP-EHH to my knowledge is appropriate for application to this kind of dataset.

We used the same XP-EHH as this publication “Sabeti, P.C. et al. Genome-wide detection and characterization of positive selection in human populations. Nature 449, 913-918 (2007).” Specifically in our analysis, the SNP information of modern maize was compared with ancient maize. The code is available in https://doi.org/10.5061/dryad.w6m905qtd.

XP-EHH is a statistical method used in population genetics to detect recent positive selection in one population compared to another, and it often applied in modern large maize populations in previous research. In our study, we wanted to detect recent positive selection in modern maize compared to ancient maize, thus, we applied XP-EHH here. Although the population size of ancient maize is not big, it is the best method that we can apply for our dataset here to detect recent selection on modern maize.

**Reviewer #3 (Public review):**
Summary:The authors seek to place archaeological maize samples (2 kernels) from Bolivia into genetic and geographical context and to assess signatures of selection. The kernels were dated to the end of the Incan empire, just prior to European colonization. Genetic data and analyses were used to characterize the distance from other ancient and modern maize samples and to predict the origin of the sample, which was discovered in a tomb near La Paz, Bolivia. Given the conquest of this region by the Incan empire, it is possible that the sample could be genetically similar to populations of maize in Peru, the center of the Incan empire. Signatures of selection in the sample could help reveal various environmental variables and cultural preferences that shaped maize genetic diversity in this region at that time.Strengths:The authors have generated substantial genetic data from these archaeological samples and have assembled a data set of published archaeological and modern maize samples that should help to place these samples in context. The samples are dated to an interesting time in the history of South America during a period of expansion of the Incan empire and just prior to European colonization. Much could be learned from even this small set of samples.Weaknesses:

Many thanks for your comments and suggestions. We have addressed these below and provided further explanation.

(1) Sample preparation and sequencing:Details of the quality of the samples, including the percentage of endogenous DNA are missing from the methods. The low percentage of mapped reads suggests endogenous DNA was low, and this would be useful to characterize more fully. Morphological assessment of the samples and comparison to morphological data from other maize varieties is also missing. It appears that the two kernels were ground separately and that DNA was isolated separately, but data were ultimately pooled across these genetically distinct individuals for analysis. Pooling would violate assumptions of downstream analysis, which included genetic comparison to single archaeological and modern individuals.

We did not do the morphological assessment of the samples and comparison to morphological data from other maize varieties because we only have 2 aBM kernels, and we do not have other archaeological samples that could be used to do comparison.

For “It appears that the two kernels were ground separately and that DNA was isolated separately, but data were ultimately pooled across these genetically distinct individuals for analysis”, as you can see from our Materials and Methods section that “Whole kernels were crushed in a mortar and pestle”, these two kernels were ground together before sequenced.

While morphological assessment of the sample would be interesting, most morphological data reported for maize are from microremains (starch, phytoliths, pollen) and this is beyond the scope of our study. Most studies of macrobotanical remains do not appear to focus solely on individual kernels, but instead on (or in combination with) cob and ear shape, which were not available in the assemblage.

(2) Genetic comparison to other samples:The authors did not meaningfully address the varying ages of the other archaeological samples and modern maize when comparing the genetic distance of their samples. The archaeological samples were as old as >5000 BP to as young as 70 BP and therefore have experienced varying extents of genetic drift from ancestral allele frequencies. For this reason, age should explicitly be included in their analysis of genetic relatedness.

We have changed related part in our revised manuscript.

(3) Assessment of selection in their ancient Bolivian sample:This analysis relied on the identification of alleles that were unique to the ancient sample and inferred selection based on a large number of unique SNPs in two genes related to internode length. This could be a technical artifact due to poor alignment of sequence data, evidence supporting pseudogenization, or within an expected range of genetic differentiation based on population structure and the age of the samples. More rigor is needed to indicate that these genetic patterns are consistent with selection. This analysis may also be affected by the pooling of the Bolivian archaeological samples.

We do not think it is because of poor alignment of sequence data since we used BWA v0.7.17 with disabled seed (-l 1024) and 0 mismatch alignment. Therefore, there are no SNPs that could come from poor alignment. Please see our detailed methods description here “For the archaeological maize samples, adapters were removed and paired reads were merged using AdapterRemoval60 with parameters --minquality 20 --minlength 30. All 5՛ thymine and 3՛ adenine residues within 5nt of the two ends were hard-masked, where deamination was most concentrated. Reads were then mapped to soft-masked B73 v5 reference genome using BWA v0.7.17 with disabled seed (-l 1024 -o 0 -E 3) and a quality control threshold (-q 20) based on the recommended parameter61 to improve ancient DNA mapping”.

For “More rigor is needed to indicate that these genetic patterns are consistent with selection”, Could you please be more specific about which method or approach we should use here? For example, methods from specific publications that could be referenced? Or which specific tool could be used?

“This analysis may also be affected by the pooling of the Bolivian archaeological samples.” As we could not prove these two seeds came from two different individual plants, we do not think this analysis was affected by the pooling of the Bolivian archaeological samples.

(4) Evidence of selection in modern vs. ancient maize: In this analysis, samples were pooled into modern and ancient samples and compared using the XP-EHH statistic. One gene related to ovule development was identified as being targeted by selection, likely during modern improvement. Once again, ancient samples span many millennia and both South, Central, and North America. These, and the modern samples included, do not represent meaningfully cohesive populations, likely explaining the extremely small number of loci differentiating the groups. This analysis is also complicated by the pooling of the Bolivian archaeological samples.

Yes, it is possible that ovule development might be a modern improvement. We re-wrote this part in our revised manuscript.

**Reviewer #1 (Recommendations for the authors):**
My suggestion is to address the comments that outline why the methods used or results obtained are not sufficient to support your conclusions. Overall, I suggest limiting the narrative of Inca influence and framing it as speculation in the discussion section. Presenting conclusions of Inca influence in the title and abstract is not appropriate, given the very questionable evidence.

We agree and have changed the title to “Fifteenth century CE Bolivian maize reveals genetic affinities with ancient Peruvian maize”.

**Reviewer #2 (Recommendations for the authors):**
(1) Line 74: Mexicana is another subspecies of teosinte; the distinction is between ssp. mexicana and ssp. parviglumis (Balsas teosinte), not mexicana and teosinte.

We have corrected this in our revised manuscript.

(2) Line 100-102: This is a bit confusing, it cannot have been a symbol of empire "since its first introduction", since its introduction long predates the formation of imperial politics in the region. Reference 17 only treats the late precolonial Inca context, while ref 22 (which cites maize cultivation at 2450 BC, not 3000 BC) makes no reference to ritual/feasting contexts; it simply documents early phytolith evidence for maize cultivation. As such, this statement is not supported by the references offered.

lines 100-102. This point is well taken and was poor prose on our part. We have modified this discussion to reflect both the confusing statement and we have corrected our mistake in age for reference 22. associated prose has been modified accordingly.

We have corrected them as “Indeed, in the Andes, previous research showed that under the Inca empire, maize was fulfilled multiple contextual roles. In some cases, it operated as a sacred crop” and “…since its first introduction to the region around 2500 BC”.

(3) Line 161: IntCal is likely not the appropriate calibration curve for this region; dates should probably be calibrated using SHCal.

We greatly appreciate this important (and correct) observation. We have completely recalibrated the maize AMS result based on the southern hemisphere calibration curve, discussed the new calibrations, and have also invoked two other AMS dates also subjected to the southern hemisphere calibration on associated material for comparison.We are confident in a 15th century AD/CE age for the maize, most likely mid- to late 15th century.

(4) Lines 167-169: The increase of G and A residues shown in Supplementary Figure S1a is just before the 5' end of the read within the reference genome context, and is related to fragmentation bias - a different process from postmortem deamination. Deamination leads to 5' C->T and 3' G->A, resulting in increased T at 5' ends and increased A at 3' ends, and the diagnostic damage curve. The reduction of C/T just before reads begin is not a result of deamination.

We have removed the “Both features are indicative of postmortem deamination patterns” in our revised manuscript.

(5) Lines 187-196 This section presents a lot of important external information establishing hypotheses, and needs some references.

We have added the related references here.

(6) Line 421: This makes it sound like damage masking was done BEFORE read mapping. However, this conflicts with the previous paragraph about map Damage, and Supplementary Figure 1 still shows a slight but perceptible damage curve, which is impossible if all terminal Ts and As are hard-masked. This should be reconciled.

The Supplementary Figure 1 shows the raw ancient maize DNA sample before damage masking. Specifically, Step1: We used map Damage to check/estimate if the damage exists, and we made the Supplementary Figure 1. Step 2: Then we used our own code hard-masked the damage bases and did read mapping.

The purpose of Supplementary Figure 1 is to show the authenticity of aBM as archaeological maize. Therefore, it should show a slight but perceptible damage curve.

(7) Line 460: PCA method is not given (just the LD pruning and the plotting).

The merged dataset of SNPs for archaeological and modern maize was used for PCA analysis by using “plink –pca”.

(8) "tropicalis" maize is not common usage, it is not clear to me what this refers to.

We have changed all “tropicalis maize” as “tropical maize” in our revised manuscript.

(9) The Figure 4 color palette is not accessible for colorblind/color-deficient vision.

We have changed the color of Figure 4. Please find the new colors in our upload Figure 4.

(10) Datasets S2 and S3 are not included with this submission.

Thank you for letting us know and your suggestion. We have included Datasets S2 and S3 here.